# Drug Utilization and Medication Adherence: A Data-Driven Analysis of Drugs with Different Routes of Administration Applied in Atopic Dermatitis

**DOI:** 10.3390/pharmaceutics17101279

**Published:** 2025-10-01

**Authors:** Sara Mucherino, Annunziata Raimondo, Milana Krstin, Ignacio Aznar-Lou, Marianna Serino, Lara Perrella, Francesca Futura Bernardi, Ugo Trama, Enrica Menditto, Serena Lembo, Valentina Orlando

**Affiliations:** 1Department of Pharmacy, University of Naples Federico II, 80131 Naples, Italyenrica.menditto@unina.it (E.M.); valentina.orlando@unina.it (V.O.); 2Center of Pharmacoeconomics and Drug Utilization Research (CIRFF), University of Naples Federico II, 80131 Naples, Italy; 3HealthCare Datalab, Campania Region, 80143 Naples, Italy; 4Department of Medicine, Surgery and Dentistry, “Scuola Medica Salernitana”, University of Salerno, 84081 Baronissi, Italy; 5Health Technology Assessment in Primary Care and Mental Health (PRISMA) Research Group, Parc Sanitari Sant Joan de Déu, Institut de Recerca Sant Joan de Déu, St Boi de Llobregat, Catalonia, 08950 Esplugues de Llobregat, Spain; 6Consortium for Biomedical Research in Epidemiology and Public Health (CIBERESP), 28029 Madrid, Spain; 7Regional Pharmaceutical Unit, Campania Region, 80143 Naples, Italy

**Keywords:** medication adherence, drug utilization, medication patterns, adherence determinants, atopic dermatitis

## Abstract

**Background:** Medication adherence is one of the critical factors in optimizing treatment outcomes for chronic diseases such as atopic dermatitis (AD). Existing studies use aggregate data, but there is a need for assessment of medication adherence phases, such as the initiation and discontinuation of therapy. The aim of this study was to assess medication adherence across patients with moderate to severe AD, investigating the impact of drug treatment characteristics, particularly the route of administration, on adherence levels during treatment. **Methods:** A retrospective observational study on an Italian sample included 821 newly diagnosed AD patients from January 2021 to June 2022. Medication adherence was evaluated by EMERGE guidelines, focusing on initiation and discontinuation. Discontinuation was assessed at 6 and 12 months, comprising sensitivity analysis. Statistical analysis included chi-square tests and descriptive statistics on treatment duration. **Results:** Treatment initiation is significantly lower for tacrolimus ointment (38% non-initiation) than for dupilumab injection (12% non-initiation), due to initial healthcare support for dupilumab patients. After six months, 75.6% of dupilumab injection patients remained on therapy, while 24.4% of patients continued tacrolimus ointment treatment. After one year, therapy persistence was 68.7% among users of dupilumab, while only 22.5% of patients remained on tacrolimus therapy. Dupilumab demonstrated a significantly longer median treatment duration compared to tacrolimus (4.4 vs. 2.6 months; *p* < 0.01). **Conclusions**: The observed differences in adherence patterns between topical tacrolimus and subcutaneous dupilumab suggest that distinct contextual and behavioral factors influence patient adherence during therapy.

## 1. Introduction

Medication adherence is a crucial factor for the optimization of the effectiveness of pharmacological treatments, especially for chronic conditions [1,2,3,4,5]. To date, it is widely recognized that suboptimal medication adherence can lead to negative clinical outcomes, increased healthcare costs, and reduced quality of life for patients [6,7,8,9,10,11]. Considering this impact, understanding and improving medication adherence is a key public health and clinical practice concern [12].

According to the EMERGE Guidelines and the ABC Taxonomy on adherence [13,14], adherence is a complex process, which includes three distinct phases: initiation of the therapy, implementation of the prescribed dosing schedule over time, and persistence or discontinuation of prescribed therapy [14]. Despite this, to date, existing research on adherence often provides aggregate data, and there is a lack of studies that stratify results into specific adherence phases. This could lead to more realistic insights into patient adherence behavior [15,16].

Hence, by investigating the specific medication adherence phases, it is possible to identify specific determinants for each phase of adherence [6]. Among these, the characteristics of the pharmaceutical product itself, including its dosage form, formulation, and administration route, are increasingly recognized as one of adherence determinants in the treatment of dermatological conditions, such as atopic dermatitis (AD) [17,18]. Hence, AD, also known as atopic eczema, is a chronic, pruritic, inflammatory dermatosis affecting up to 25% of children and 2% to 3% of adults worldwide, with a prevalence of 8.1% in Italy [19,20]. Conventional treatments, such as topical corticosteroids and immunosuppressants, while effective, often come with limitations and potential side effects that can affect patient acceptance and long-term use [21]. The innovative therapies for AD, such as biologics (e.g., dupilumab, tralokinumab, and others), emphasize the need to understand patients’ behaviors during treatment, as these agents are administered via subcutaneous injection, which differs from other conventional options such as topical or oral treatments [22,23,24,25].

Despite the growing field of research on medication adherence in several chronic diseases, specific data on its phases in AD and its determinants remain limited [26,27,28,29]. Hence, understanding these stages in dermatological chronic conditions is crucial for identifying barriers to adherence and tailoring interventions to improve therapeutic outcomes. To date, there are a few studies investigating the influence of pharmaceutical formulations in dermatological conditions, particularly for AD treatments, on adherence outcomes [17,18]. Hence, it was already recognized that patient-centered pharmaceutical drug product design offers a strategic approach to addressing this gap by designing medicinal products that align with patients’ needs and preferences [30].

The primary aim of this study was to assess medication adherence across patients with moderate to severe AD, investigating the impact of drug treatment characteristics, particularly the route of administration, on adherence levels during treatment.

## 2. Materials and Methods

### 2.1. Study Design, Population, and Data Sources

A retrospective observational study was carried out in the Campania Region, a Southern Italian Region of about 6 million inhabitants (10% nationality representation). The Italian healthcare system, including that of Campania, is organized around universal healthcare principles, primarily funded through taxation and managed at the regional level, with variations in care delivery across regions. As in all other Italian regions, healthcare services are provided to all citizens and legal foreign residents through Local Health Units (LHUs). About 99% of them are covered by the public healthcare system.

This study is reported according to the Strengthening of the Reporting of Observational Studies in Epidemiology guidelines [31]. The data source was the Collection of Treatment Plans of the Campania Region. This data collection system operates as a web platform for specialized facilities and public and private pharmacies within the Local Health Units (LHUs) of the Campania Region. It includes treatment plans that provide information on diagnosis (coded with ICD-9-CM) and medication details (coded with the Anatomical Therapeutic Chemical, or ATC) for the active ingredients prescribed in the form of reimbursement. We merged this database with the Campania Region DataBase (CaReDB), an electronic health data warehouse previously validated in several studies [32,33,34,35]. CaReDB provided demographic information on all residents covered by the Regional Health System, including age and sex. To protect patient privacy, these two databases were linked through a unique, encrypted, and anonymous identifier.

Patients were enrolled from 1 January 2021 to 30 June 2022 if they had an incident diagnosis of AD (no diagnosis registry of AD in the previous 3 years) with the following diagnosis codes: ICD-9-CM codes 691.0 and 691.8. The entire study design is depicted in Appendix A. Patients with moderate to severe AD were selected, identified as those who received prescriptions for approved drugs for this disease’s severity (see Appendix A). The first drug prescription date was considered as the index date, and patients were followed for one year from that date. Hence, the final study cohort included moderate to severe AD patients who started treatment with dupilumab (subcutaneous injection formulation), tacrolimus (topical formulation), or ciclosporin (oral formulation), to address medication adherence in different routes of administration.

### 2.2. Adherence Measurement

Medication adherence was assessed by calculating its different phases, according to the EMERGE guidelines on adherence [13,14,36], comparing pharmacological treatments with different routes of administration, prescribed to a cohort of chronic patients diagnosed with moderate to severe atopic dermatitis (AD). This evaluation included topical tacrolimus and injectable dupilumab, as the cohort of oral cyclosporine recorded a sample size that was not statistically significant. Although administered for the same chronic condition, tacrolimus ointment and subcutaneous dupilumab differ substantially in terms of their regulatory classification, dispensing setting, prescriber type, and clinical indication within the treatment pathway (details are shown in Appendix A) [37,38,39]. Tacrolimus ointment is a Class A medication (reimbursed by the Italian healthcare system), dispensed through community pharmacies and prescribed by specialists, typically dermatologists. It is used as a topical anti-inflammatory agent, often in earlier stages of treatment, or as maintenance therapy [37]. Dupilumab, on the other hand, is a Class H hospital-only medication, dispensed through hospital pharmacies and also prescribed by dermatology specialists, but typically in the hospital setting. It is indicated for patients eligible for systemic therapy, often due to severe or refractory disease, and its initiation usually follows a structured eligibility assessment [38]. The two treatments also differ in terms of route of administration. Hence, tacrolimus requires twice-daily topical application, while dupilumab is administered as a subcutaneous injection every other week. These contextual differences may influence medication adherence behaviors, particularly in relation to treatment initiation and persistence in real-world clinical practice. Therefore, this research has a specific focus on these two phases of the adherence process: initiation and persistence (discontinuation). Initiation, reflecting primary non-adherence [13,14,36,40]. Specifically, a patient was classified as exhibiting primary non-adherence if, after receiving an initial prescription for AD treatment, no subsequent prescriptions for the same medication were recorded. Such individuals were categorized as spot patients, indicating that therapy was not effectively initiated. Otherwise, discontinuation was defined as the interruption of therapy, computerized as the absence of any further prescriptions (identified as refills) for at least 12 months following the last recorded prescription. The date of discontinuation was set as the date of the last medication refill recorded. For the calculation of discontinuation, drug-specific coverage periods were calculated according to their suggested posology, as detailed in the approved summary of product characteristics (Appendix A). The coverage period was calculated by considering the prescribed dosage regimen, the quantity dispensed (e.g., the number of packages for oral therapies and volume in milliliters for topical therapies), and a grace period. The grace period was adjusted using sensitivity analyses to account for variations in prescribing and consumption patterns. To measure the proportion of patients remaining on therapy, adherence was calculated monthly as the ratio of patients who had not discontinued therapy to the total number of patients who had initiated treatment. This approach provided a dynamic evaluation of treatment persistence over the follow-up period, incorporating variations in drug utilization patterns.

### 2.3. Statistical Analyses

Descriptive and inferential statistical methods were carried out to analyze medication adherence to different AD treatments. Continuous variables were expressed as means and standard deviations (SDs), while categorical variables were presented as absolute numbers and percentages. Moreover, comparisons between groups were performed using the chi-square test for categorical variables to assess the statistical significance of differences between proportions. A *p*-value threshold of <0.05 was considered to be statistically significant. Otherwise, for continuous variables, differences in treatment durations across therapies were analyzed using independent *t*-tests or analysis of variance with the ANOVA test, as appropriate. For the application of the *t*-tests and ANOVA tests, the normality of data distribution was assessed using the Shapiro–Wilk test and Q–Q plots, while homogeneity of variances was verified using Levene’s test. For adherence metrics, initiation rates were calculated as the proportion of patients who filled at least one subsequent prescription (refill medication) after the first. Discontinuation rates were evaluated at 6 and 12 months, considering periods of drug coverage and incorporating sensitivity analyses. Boxplots were employed to illustrate the distribution of months to discontinuation for the overall cohort and stratified by treatment groups. Statistical analyses and visualizations were performed using R statistical software (version R-4.4.2).

## 3. Results

A total of 2036 patients with a diagnosis of AD were identified as prevalent cases during the study period (Appendix A). Among those, 825 were classified as incident AD cases. Of the incident cohort, four patients started an oral treatment with cyclosporine, and, due to this sample size (not significant), this sub-cohort of patients was not included in the medication adherence analyses. As a result, the final AD cohort comprised 821 incident patients who initiated treatment with either an injectable subcutaneous formulation with dupilumab (*n* = 661, 80.5%) and a topical formulation of tacrolimus ointment (*n* = 160, 19.5%) (Table 1). The sex distribution was mainly balanced across groups (females: 49.7%; males: 50.3%), with no significant difference between treatment routes of administration (*p* = 0.065). However, patients receiving dupilumab were significantly older compared to those receiving tacrolimus (mean age: 38.9 ± 20.9 vs. 30.2 ± 21.6 years, respectively; *p* < 0.01). This age difference is mainly driven by a higher proportion of pediatric patients (6–11 years old) prescribed tacrolimus (23.8%) compared to subcutaneous injection of dupilumab (0.3%). Conversely, dupilumab was predominantly prescribed among adult age groups (18–64 years: 67.9%; ≥65 years: 16.0%) (Table 1).

### 3.1. Adherence Assessment in the Initiation Phase

Medication adherence during the initiation phase (primary adherence) is detailed in Table 2 and Figure 1. Overall, primary non-adherence was observed in 17.1% of patients in the AD cohort. Patients who started therapy with topical tacrolimus ointment showed significantly higher primary non-adherence rates (38.1%) compared to those receiving subcutaneous dupilumab injections (12%). Sex differences were revealed, particularly among patients treated with tacrolimus ointment, where females showed notably higher primary non-adherence (57.4%) compared to males (42.6%). Figure 1 stratifies primary non-adherence rates by age group, clearly demonstrating higher non-adherence among pediatric patients (aged 6–11 years, 34.4%) who started treatment with tacrolimus ointment. Conversely, significant differences were observed for adults (18–64 years) and older adults (older than 65 years) with the highest primary non-adherence when treated with subcutaneous dupilumab injections (older adults: 25.3% dupilumab vs. 9.8% tacrolimus, *p*-value 0.03).

### 3.2. Adherence Assessment in the Discontinuation Phase

Adherence results in the discontinuation phase at 6 and 12 months are shown in Table 2 and Figure 2. Within 6 months, 34.3% of patients had discontinued treatment, with a significantly higher proportion for tacrolimus ointment patients (75.6%) compared to subcutaneously injected dupilumab (24.4%; *p* < 0.001). Conversely, within 12 months from the start of the treatment, overall discontinuation increased to 40.3%, again significantly higher among tacrolimus-treated patients (77.5%) compared to dupilumab-treated patients (31.3%; *p* < 0.001). The mean duration of therapy before discontinuation was considerably shorter for tacrolimus ointment (117.4 ± 142.4 days) than for dupilumab (278.7 ± 141.5 days). Figure 2 confirms these findings, illustrating significantly shorter persistence and higher early discontinuation rates among tacrolimus-treated patients.

## 4. Discussion

This is the first study investigating medication adherence to AD treatments, reporting different adherence behaviors during the initiation and the discontinuation of treatment, influenced by the route of administration. Hence, this study addressed significant differences in the medication use and adherence outcomes in real practice when using injections or topical applications of AD treatments. Guided by the EMERGE adherence reporting guidelines [13] and the ABC taxonomy framework [14], this analysis evaluated patients who initiated therapy, revealing that 38.1% of those prescribed topical treatment of tacrolimus ointment did not dispense their medication from the pharmacy. In contrast, only 12% of patients prescribed injection treatment, specifically subcutaneous dupilumab, did not initiate treatment. Nevertheless, persistence with subcutaneous dupilumab treatment remained high throughout the entire treatment period. Furthermore, persistence with tacrolimus ointment was significantly lower; only 24.4% of patients continued treatment after the first six months, and just 22.5% remained persistent within a year.

Adherence determinants, that may reflect patient preferences and challenges associated with topical formulations, include a high frequency of application [41], the absence of a fixed therapeutic plan [42], and a lack of structured therapeutic monitoring by healthcare professionals [43]. Moreover, the use of a specific topical vehicle with differences in the texture (e.g., ointment, typically greasy and occlusive, versus hydrogels, water-based and more breathable) could influence medication adherence, resulting in different adherence outcomes [44]. Regarding the high frequency of application, it is noteworthy that tacrolimus ointment is prescribed twice a day to moderate-to-severe AD patients, which requires careful local application [37,45,46]. This high frequency of administration can be a challenge for patients, reducing their adherence to therapy [47]. Patients may discontinue treatment due to side effects like skin burning and pruritus, which are most common when starting the medication and tend to decrease over time. This can lead the patient to underuse the medication, resulting in a negative impact on its effectiveness [47]. In contrast, dupilumab is administered once every two weeks, following the initial dose, with mandatory initial training in proper administration [37]. Although patients self-administer this therapy, less frequent dosing with clearly defined education may contribute to a patient’s highest persistence in using the dupilumab subcutaneous injections [38]. Another hypothesis is that tacrolimus does not have a fixed therapeutic schedule, as patients were instructed by the prescriber to topically administer it based on the patient’s clinical response [39]. Indeed, according to clinical guidelines, after achieving improvement, tacrolimus should be self-administered intermittently or as needed, within a reactive or proactive therapeutic approach [39]. While this approach is clinically justified, it may result in lower medication persistence in real-world practice; hence, in this study, only 22.5% of AD patients continued treatment within a year. On the other hand, the dupilumab treatment regimen is fixed as it includes a precisely defined dosage and administration schedule, and prescription renewal is possible only through the hospital pharmacist [30,38,42,48]. Moreover, another explanation could be related to the lack of specific monitoring requirements for tacrolimus ointment administration, despite the need for caution in long-term use, which may negatively influence treatment persistence [43,49]. Differently, dupilumab adherence can decrease over time, with reasons for loss of adherence including lack of efficacy and adverse effects. While generally well-tolerated, some adverse events, such as conjunctivitis, psoriasiform dermatitis, and arthralgia, can lead to treatment withdrawal. Some patients choose to stop the treatment course for other reasons, which can include achieving remission, pregnancy, or other life events [38,50]. Based on the main findings of this study, the role of the pharmaceutical administration route appears to be significant in supporting medication adherence, particularly from a patient-centered perspective. This is in line with the current literature [17,51]. Indeed, despite topical formulations being the most commonly used treatment for dermatological conditions, they often result in lower levels of patient adherence, as has already been reported in the literature [44,51,52]. Hence, the study by Saeki et al. reports that oral administration of the treatment to dermatological patients is associated with higher adherence rates compared to topical agents [53]. This finding supports the hypothesis that the routine of administration may significantly influence medication adherence in chronic conditions. Indeed, this association has also been observed in other chronic diseases, indicating that patients receiving injectable medications often exhibit higher adherence to initial treatment compared to those receiving oral formulations [54,55,56,57].

Regarding drug utilization patterns of patients diagnosed with moderate-to-severe AD, these findings also reflect that injection with dupilumab was the most prescribed in the Italian sample, independent of their sex or age, in comparison with the two alternatives, topical tacrolimus and oral cyclosporine. Dupilumab injection shows higher rates of adherence at all time points in comparison with tacrolimus ointment. The increasing predominance of dupilumab injection in the treatment of AD in recent years has already been reported previously in other contexts since its commercialization [58]. A French qualitative study concludes that patients using dupilumab for AD reported ease of use and satisfaction [59]. It has been observed that both initiation and implementation phases of adherence are unsatisfactory when the medication is topical, in comparison with other pharmacological forms, as patients report issues with the formulation, absorption, or ease of application, which could impact implementation [60].

Several real-world studies support high persistence during dupilumab treatment in AD, which aligns with our findings. This is the case of the study by Silverberg et al., which evaluated treatment persistence rates with dupilumab, showing results comparable to our sample (91.9% vs. 75.6% at 6 months and 77% vs. 68.7% at 12 months, respectively) [26]. These rates were also confirmed from the study by Sanfilippo et al., reporting that 86% of dupilumab-treated patients were persistent [61]. Furthermore, the literature shows that the most common reasons for discontinuing dupilumab treatment could be related to insufficient effectiveness, side effects, or disease progression [26,61,62]. Dupilumab can cause adverse effects related to the systemic immune response, including rare allergic reactions such as hypersensitivity reactions, anaphylactic reactions, and serum sickness [38]. On the other hand, the adverse effects of tacrolimus ointment are mainly local and include skin irritation at the site of application, a burning sensation and itching, and erythema and redness of the skin [37].

Findings of this study reinforce the role of demographic factors, such as age and sex, as determinants in adherence patterns in AD [63]. The observed age differences between treatment groups likely reflect real-world prescribing behaviors influenced by regulatory timelines. Dupilumab’s initial approval for adult patients with severe AD, followed by later indications for pediatric use, may have contributed to the older age profile observed among users of this treatment [64]. Furthermore, tacrolimus ointment has been more commonly prescribed to the pediatric population, as is indicated for use in patients over two years of age, whereas dupilumab has been approved for use in children older than six months.

Moreover, these findings partially diverge from previous studies regarding sex-related differences in AD pharmaceutical treatment [65,66,67]. A multicenter Italian study found no significant differences in disease severity or burden between male and female patients with AD [68], whereas our results suggest slightly better adherence among males. Similarly, a recent study investigating barriers to dupilumab self-injection in patients with severe allergic diseases reported only a non-significant trend toward higher adherence in males. In that study, the only factor significantly associated with adherence was the duration of dupilumab therapy [69].

Another crucial consideration is that clinical guidelines suggest prescribing topical medications for moderate-to-severe AD as the first- or second-line treatment [39]. However, it is demonstrated that these topical applications are associated with lower patient persistence to treatment [39,70,71,72], and this could cause symptoms to worsen, subsequently leading to the initiation of systemic therapy [39]. The Italian healthcare system mainly covers the costs of these treatments for moderate to severe cases; therefore, the worsening of the disease and the patient’s transition to third-line treatment and thus to the introduction of systemic therapy could represent a significant additional financial burden for the healthcare system, considering the considerable difference in cost for systemic ADs (Appendix A). Moreover, apart from the direct drug-related healthcare costs, this therapeutic approach additionally burdens the health system through an increased need for specialist supervision, complex administration, and greater involvement of healthcare providers, which, in general, leads to an increase in organizational pressure on healthcare resources [73].

This study has several limitations to address. It provides evidence on underexplored aspects such as medication adherence in AD and the use of a specific routine of administration, offering insights into the adherence process and its phases. Also, drugs belonging to the category of JAK inhibitors were not included in the analyses as they have been on the market in Italy since 2023, after the end of the study period. However, the focus of this study was to provide insight into the correlation between patient adherence behavior and different drugs with diverse routes of administration, which could also be considered in new therapeutic options. Moreover, the study compares two administration routes, among others. Although the oral formulation was not analyzed due to a small sample size, the study’s sample, representative of one of the most populated regions in Italy, provides robust support for these conclusions. Furthermore, although clinical data were not available for this study (such as information on clinical outcomes like disease progression, remission, or adverse drug reactions), using administrative prescription data alone allowed for an accurate assessment of medication adherence in real-world routine practice. However, this approach does not allow for the determination of the exact reasons for treatment discontinuation. Additionally, the data set comprises issued prescriptions and does not include information about the dispensing of the medication at the pharmacy, so we assumed that all issued prescriptions were filled. This lack of information could affect adherence rates for both study drugs, due to patients’ beliefs about the disease or the medication’s posology in the absence of prior experience, their emotional reactions to the treatment, or their level of health literacy. Finally, this is the first study to evaluate the relationship between drug patterns and their administration route with medication adherence levels by distinguishing between its phases in a chronic dermatological condition. Hence, these findings could support the development of tailored interventions aimed at improving medication adherence, particularly by enhancing patient awareness and optimizing disease management in the early stages of treatment. Promoting adherence from the outset may lead to better health outcomes and, ultimately, reduce the overall burden and costs for healthcare systems. Moreover, these findings can help to address future research by highlighting which types of formulations are most effective for different patient groups, thereby preventing failure in the treatment’s efficacy and reducing the economic burden for healthcare systems [72,73,74,75,76].

## 5. Conclusions

This study highlights the importance of assessing and understanding medication adherence across its different phases, particularly initiation and persistence, when evaluating treatments for chronic conditions such as moderate to severe atopic dermatitis (AD). The observed differences in adherence patterns between topical tacrolimus and subcutaneous dupilumab suggest that distinct contextual and behavioral factors influence patient adherence during therapy. Therefore, interventions aimed at improving adherence should not adopt a one-size-fits-all approach. Instead, they should be tailored to the specific characteristics of each therapy.

Optimal medication adherence to treatment has the potential to prevent disease progression and reduce the need to introduce more expensive therapeutic options. Timely recognition and correction of low medication adherence not only contribute to a better treatment outcome but also significantly reduce costs for the healthcare system.

These findings are particularly significant in the treatment of chronic diseases, where patients often face the challenge of medication adherence. Hence, future research could more extensively explore the relationship between pharmaceutical formulations and medication adherence to inform health policy and clinical strategies, integrating differentiated approaches to optimize medication adherence, improve health outcomes, and reduce unnecessary healthcare costs.

## Figures and Tables

**Figure 1 pharmaceutics-17-01279-f001:**
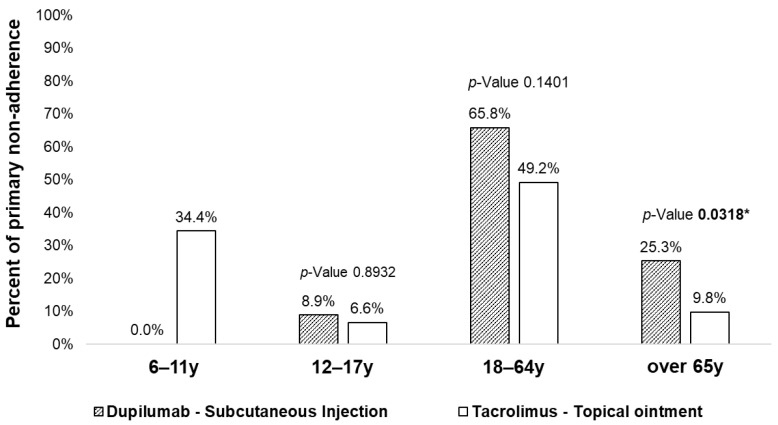
Primary non-adherence to moderate-to-severe AD treatments by pharmacological treatment and age group. Notes: the chi-square test was performed to compare the differences between the two proportions. * The *p*-value is considered statistically significant if ≤0.05. The *p*-value is not reported for comparisons where data are not available (i.e., for the 6–11 years age group, where no patients treated with dupilumab exhibited primary non-adherence).

**Figure 2 pharmaceutics-17-01279-f002:**
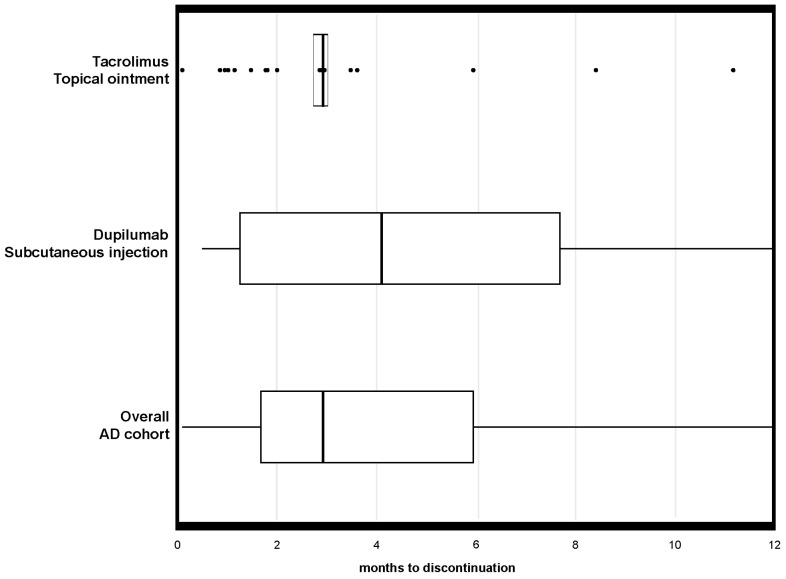
Boxplots of months to treatment discontinuation to moderate-to-severe AD by pharmacological treatments. Notes: the box represents the interquartile range (IQR) with the horizontal line indicating the median value, while the whiskers extend to 1.5 times the IQR beyond the first and third quartiles. Specifically, the boxplot displays the distribution of months to treatment discontinuation within a 12-month follow-up period and includes only patients who initiated therapy, excluding those classified as “spot/non-initiators” in the initiation phase. As such, the plot represents real persistence patterns among patients who started treatment, both for the overall cohort and stratified by administration route of the pharmacological treatment (systemic, namely subcutaneous injection vs. topical, namely topical ointment). The box for the tacrolimus group appears notably small due to the highly concentrated distribution of discontinuation times. Most patients discontinued tacrolimus ointment treatment shortly after initiation, resulting in minimal variability in treatment duration. This limited dispersion produces a compressed box, reflecting overall poor persistence in the tacrolimus-treated subgroup.

**Table 1 pharmaceutics-17-01279-t001:** Baseline characteristics by pharmacological treatment for moderate-to-severe AD.

Baseline Cohort Characteristics	AD Cohort	Dupilumab Subcutaneous Injection	Tacrolimus Topical Ointment	*p*-Value *
Overall, *n* (%)	821 °	661 (80.5)	160 (19.5)	-
Sex, *n* (%)				0.065
Female	408 (49.7)	318 (48.1)	90 (56.3)	-
Male	413 (50.3)	343 (51.9)	70 (43.8)	-
Mean age ± SD	37.2 ± 21.3	38.9 ± 20.9	30.2 ± 21.6	<0.01
Age groups, *n* (%)				
6–11 years	40 (4.9)	2 (0.3)	38 (23.8)	-
12–17 years	123 (15)	104 (15.7)	19 (11.9)	-
18–64 years	534 (65)	449 (67.9)	85 (53.1)	-
≥65 years	124 (15.1)	106 (16.0)	18 (11.3)	-

* The *p*-value is considered statistically significant if ≤ 0.05. ° Patients treated with oral formulation of cyclosporine (*n* = 4) were not considered in the analysis due to sample size.

**Table 2 pharmaceutics-17-01279-t002:** Medication adherence levels to moderate-to-severe AD pharmacological treatments.

1-Year Adherence Estimation	Moderate-to-Severe AD Cohort			Dupilumab Subcutaneous Injection			Tacrolimus Topical Ointment		
	Overall	Females	Males	Overall	Females	Males	Overall	Females	Males
Total cohort, *n* (%)	821	408 (49.7)	413 (50.3)	661	318 (48.1)	343 (51.9)	160	90 (56.3)	70 (43.8)
Primary non-adherence (non-initiation), *n* (%)	140 (17.1)	77 (55)	63 (45)	79 (12)	42 (53.2)	37 (46.8)	61 (38.1)	35 (57.4)	26 (42.6)
Discontinuation within 6 months, *n* (%)	282 (34.3)	156 (55.3)	126 (44.7)	161 (24.4)	84 (52.2)	77 (47.8)	121 (75.6)	72 (59.5)	49 (40.5)
Mean days (± SD) to discontinuation within 6 months *	133.70 ± 73.85	129.45 ± 74.86	137.90 ± 72.68	147.90 ± 67.00	146.36 ± 67.57	149.33 ± 66.53	75.04 ± 72.04	69.72 ± 68.81	81.87 ± 75.94
Discontinuation within 12 months, *n* (%)	331 (40.3)	177 (53.5)	154 (46.5)	207 (31.3)	103 (49.8)	104 (50.2)	124 (77.5)	74 (59.7)	50 (40.3)
Mean days (± SD) to discontinuation within 12 months *	247.23 ± 155.32	236.94 ± 157.85	257.40 ± 152.29	278.66 ± 141.47	274.62 ± 143.51	282.39 ± 139.66	117.42 ± 142.41	103.82 ± 132.68	134.92 ± 153.22

Abbreviations: SD, standard deviation. * The maximum number of days is 183 in 6 months and 365 in 12 months.

## Data Availability

All data used for the current study are available upon reasonable request to the Centro di Ricerca in Farmacoeconomia e Farmacoutilizzazione (CIRFF) authorized by the governance board of Unità del Farmaco della Regione Campania [D.G.R. 276, 23 May 2017]. The data are not publicly available due to legal/ethical restrictions (GDPR—Reg. (EU) 2016/679). However, aggregate results and summary tables supporting the main findings are provided within the paper and Appendix A.

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
