# Peer review of "Drug Utilization and Medication Adherence: A Data-Driven Analysis of Drugs with Different Routes of Administration Applied in Atopic Dermatitis"

_pharmaceutics, 2025, doi:10.3390/pharmaceutics17101279_

Round 1

Reviewer 1 Report

Comments and Suggestions for Authors

The manuscript entitled “Impact of pharmaceutical formulations on medication adherence: A data-driven method of drug utilization applied in atopic dermatitis” represents a comparative study of the patients adherence to two different therapeutic regimens for the management of atopic dermatitis.

In my opinion the study is not interesting for the pharmacetutics journal readers, additionally the comparison is not valid, there is no point in comparing two totally different treatment options (topical immunosuppressant tacrolimus cream against dupilomub injection).

One important factor in the continuation or adherence to medication is related to the severity of the condition and this was something totally missed by the authors. 80% of the sample size was prescribed the injection, which means that these patients had severe symptoms during diagnosis. Authors on the other side, claim that the higher adherence in case of injections was due to the nature of the treatment which makes the patient start the treatment in the clinics while it is totally known in the medical field that most patients would prefer topical treatment versus the injection due to ease of use.

Authors also claim that the lower adherence in case of topical treatment was due to the perceived less efficacy and they did not consider at all the inverse possibility of symptoms improvement that led to the discontinuation as patients feel they no more need treatments.

Additionally the discussion is very poor and lacks the consideration of many clinical and patient related factors such as the appearance of side effects, allergic reactions or even shifting to other medications.

Author Response

Response to Reviewer 1 Comments
1. Summary
The manuscript entitled “Impact of pharmaceutical formulations on medication adherence: A data-driven method of drug utilization applied in atopic dermatitis” represents a comparative study of the patients adherence to two different therapeutic regimens for the management of atopic dermatitis.
Dear Reviewer, on behalf of all co-authors, we would like to thank you for your valuable comments and suggestions. We have carefully addressed them and implemented the necessary changes in this revised version of the manuscript. Please find our detailed responses below, along with the corresponding revisions and corrections highlighted using track changes in the resubmitted files.
2. Point-by-point response to Comments and Suggestions for Authors
Comment 1: In my opinion the study is not interesting for the pharmacetutics journal readers, additionally the comparison is not valid, there is no point in comparing two totally different treatment options (topical immunosuppressant tacrolimus cream against dupilomub injection). Response 1: We totally understand differences when comparing two different drugs such as dupilumab and tacrolimus, considering that they substantially differ in their active substances, pharmacological groups, mechanisms of action, indications, pharmaceutical formulations, routes of administration, and dosage regimens. The main objective of our study, however, was not to compare the two drugs per se, but to evaluate how differences in treatment characteristics, particularly the route of administration, can influence medication adherence across different phases in a real-world cohort of patients with moderate-to-severe atopic dermatitis. Patients included in our analysis were newly treated for moderate-to-severe AD, and tacrolimus and dupilumab emerged as the two most prescribed therapies in this population. Oral cyclosporine, although indicated, did not reach a sufficient sample size for inclusion. The analysis therefore focused on comparing adherence to two treatments that differ in route of administration (topical versus subcutaneous injection) while acknowledging their broader clinical differences.
In response to this comment, we have revised the manuscript title and clarified throughout the text that the comparison involves different therapies with distinct pharmacological profiles. We also included a supplementary table (Table S2) to explicitly present the key contextual differences between the two treatments. This clarification strengthens the rationale for our approach and supports the interpretation of adherence patterns considering real-world prescribing practices. Modifications have been made throughout the text accordingly.
Comment 2: One important factor in the continuation or adherence to medication is related to the severity of the condition and this was something totally missed by the authors. 80% of the sample size was prescribed the injection, which means that these patients had severe symptoms during diagnosis.
Response 2:
We understand this important observation and we added all these aspects accordingly. Indeed, in the revised version of the manuscript, we have thoroughly restructured and expanded the discussion section to specifically address how disease severity may influence treatment choices and, consequently, medication adherence behaviors. We clarified that the high proportion of patients prescribed dupilumab likely reflects clinical decisions driven by more severe cases of AD, as per treatment guidelines and prescribing criteria. These contextual elements have now been explicitly discussed and appropriately supported by updated references. Please, consider this new
revised version of the manuscript now submitted.
Comment 3: Authors on the other side, claim that the higher adherence in case of injections was due to the nature of the treatment which makes the patient start the treatment in the clinics while it is totally known in the medical field that most patients would prefer topical treatment versus the injection due to ease of use. Response 3: We agree that topical therapies are often perceived as more convenient and less invasive compared to injectable treatments (also confirmed by literature and now better addressed in this revised manuscript version). However, real-world medication adherence is a multifactorial phenomenon that does not always align with patients’ initial preferences. The data suggests that, in the context of chronic diseases such as moderate-to-severe atopic dermatitis, the initiation and continuation of therapy may be influenced by a combination of factors, including clinical severity, the treatment setting, prescriber oversight, and perceived therapeutic benefit.
As now described in Section 2.2 of this revised version, we addressed and further discussed that our findings showed a higher initiation and persistence rate for dupilumab compared to tacrolimus. Specifically, only 12% of patients prescribed dupilumab did not initiate the therapy, whereas 38.1% of those prescribed tacrolimus ointment failed to initiate treatment (table 2). Moreover, persistence after 12 months was significantly higher for dupilumab (52.4%) compared to tacrolimus (22.5%).
These results may be partly explained by the structured care pathway associated with dupilumab administration. Considering the Italian clinical guidelines (SIDeMaST guidelines and the summary of product characteristics), the first injection of dupilumab is usually administered under the supervision of a healthcare provider, with patient training for self-injection. This initial contact may foster both confidence and accountability, supporting better adherence trajectories. In contrast, tacrolimus is a topical medication typically applied without direct clinical supervision. Its twice-daily application, lack of perceived efficacy if used incorrectly, and potential for skin irritation may contribute to early discontinuation. Therefore, while we recognize that topical treatments might be preferred in terms of comfort or familiarity, our real-world analysis revealed that, in practice, injectable biologics like dupilumab were associated with better adherence outcomes, possibly due to the structured, specialist-driven treatment pathway and clearer expectations of efficacy. We addressed all these aspects in this new submitted version of the manuscript.
Comment 4: Authors also claim that the lower adherence in case of topical treatment was due to the perceived less efficacy and they did not consider at all the inverse possibility of symptoms improvement that led to the discontinuation as patients feel they no more need treatments. Response 4: We fully agree that multiple, sometimes opposing, factors can influence discontinuation of therapy in real-world settings, including both perceived lack of efficacy and early symptom relief leading to treatment discontinuation. This important aspect was not sufficiently addressed in the first version of the manuscript and we provided to address it in this new revised version. Accordingly, in this revised manuscript version, we have expanded the discussion section to explicitly acknowledge all plausible explanations for lower adherence observed in the tacrolimus cohort, as suggested. Comment 5: Additionally the discussion is very poor and lacks the consideration of many clinical and patient related factors such as the appearance of side effects, allergic reactions or even shifting to other medications. Response 5: We have completely revised the discussion section and integrated the consideration
of several clinical and patient-related factors, including the occurrence of side effects, allergic reactions, and potential treatment switching.

Reviewer 2 Report

Comments and Suggestions for Authors

This manuscript explores treatment adherence using a less-used but rationale framework of a phased approach.

I offer the following observations for the authors:

Major points:

  • The difference cannot be completely credited to the difference between formulations. The title gives the feel that the difference may be due to different formulations of the same drug (implied meaning), whereas this is an example of different active drugs. The title (and corresponding text) may be changed to something like “Impact of differences in drugs on medication adherence….”
  • Lines 140-148 – The authors can mention if they checked for the data distribution and other assumptions required for t-test, ANOVA etc.
  • The authors may specify whether the whiskers of the boxplot in Figure 2 refers to interquartile range or some other measure of dispersion
  • The last paragraph of the conclusion section provides some future steps that may be better suited to the corresponding position of the discussion section

Minor point:

  • I am not sure if the authors can do this now but it would be interesting to see a regression model with treatment response as a predictor, so we can see how the different adherence patterns can be explained by (or persistent even after controlling for)

Author Response

Response to Reviewer 2 Comments
1. Summary
This manuscript explores treatment adherence using a less-used but rationale framework of a phased approach.
Dear Reviewer, on behalf of all co-authors, we thank you for the valuable comments which we implemented in this new revised version of the manuscript. Please find detailed responses below and the corresponding revisions/corrections in track changes in the resubmitted files.
2. Point-by-point response to Comments and Suggestions for Authors
Comment 1: The difference cannot be completely credited to the difference between formulations. The title gives the feel that the difference may be due to different formulations of the same drug (implied meaning), whereas this is an example of different active drugs. Response 1: We totally understand differences when comparing two different drugs such as dupilumab and tacrolimus, considering that they substantially differ in their active substances, pharmacological groups, mechanisms of action, indications, pharmaceutical formulations, routes of administration, and dosage regimens. The main objective of our study, however, was not to compare the two drugs per se, but to evaluate how differences in treatment characteristics, particularly the route of administration, can influence medication adherence across different phases in a real-world cohort of patients with moderate-to-severe atopic dermatitis. Patients included in our analysis were newly treated for moderate-to-severe AD, and tacrolimus and dupilumab emerged as the two most prescribed therapies in this population. Oral cyclosporine, although indicated, did not reach a sufficient sample size for inclusion. The analysis therefore focused on comparing adherence to two treatments that differ in route of administration (topical versus subcutaneous injection) while acknowledging their broader clinical differences.
In response to this comment, we have revised the manuscript title and clarified throughout the text that the comparison involves different therapies with distinct pharmacological profiles. We also included a supplementary table (Table S2) to explicitly present the key contextual differences between the two treatments. This clarification strengthens the rationale for our approach and supports the interpretation of adherence patterns considering real-world prescribing practices. Modifications have been made throughout the text accordingly.
Comment 2: The title (and corresponding text) may be changed to something like “Impact of differences in drugs on medication adherence….”
Response 2: Considering the previous answer, we also agree that the “route of administration” is more pertinent in our comparative analysis. In response to this suggestion, we have revised the title and carefully reviewed the entire manuscript to ensure greater clarity and consistency. All relevant details have been revised accordingly, and a new supplementary tableS2 has been added to summarize differences.
Comment 3: Lines 140-148 – The authors can mention if they checked for the data distribution and other assumptions required for t-test, ANOVA etc.
Response 3: Thanks for this suggestion. Yes, the validity of parametric tests such as the t-test and ANOVA required the verification of specific assumptions. Hence, we confirm that we assessed the distribution of continuous variables using both graphical methods (histograms, Q-Q plots) and formal statistical tests (Shapiro-Wilk test). Homogeneity of variances was evaluated using
Levene’s test. These preliminary checks confirmed that the assumptions of normality and equal variances were reasonably satisfied for the variables compared. We now addressed these in the new revised version of the manuscript in the materials and methods section paragraph 2.3. Statistical Analyses.
Comment 4: The authors may specify whether the whiskers of the boxplot in Figure 2 refers to interquartile range or some other measure of dispersion
Response 4: We confirm that the whiskers in Figure 2 represent 1.5 times the interquartile range (IQR) from the lower and upper quartiles. This is the default definition used in the construction of boxplots with the ggplot2 package in R, which was employed for our analysis and figure generation. We have now added this specification to the figure legend to clarify this aspect for readers.
Comment 5: The last paragraph of the conclusion section provides some future steps that may be better suited to the corresponding position of the discussion section
Response 5: Please note that we completely revised the conclusion section as appropriate.
Comment 6: I am not sure if the authors can do this now but it would be interesting to see a regression model with treatment response as a predictor, so we can see how the different adherence patterns can be explained by (or persistent even after controlling for)
Response 6: Many thanks for this valuable suggestion. However, we acknowledge that treatment response data are not available in our current dataset, which is based on prescription claims. In the absence of clinical outcome measures such as EASI scores, Investigator Global Assessment (IGA), or other validated indicators of disease severity or improvement, we preferred to not infer treatment response indirectly or construct regression models that might overinterpret adherence patterns. Should clinical data become available in future research, we fully agree that examining the relationship between adherence trajectories and treatment effectiveness would be highly informative and worthy of investigation.

Reviewer 3 Report

Comments and Suggestions for Authors

Interesting approach using a data-driven method to compare adherence between two distinct treatment plans for atopic dermatitis (AD) patients.

While the main impact qualifier is described as “pharmaceutical formulation,” it is actually the route of administration (topical vs. subcutaneous injection) that plays a more significant role.

To fully understand the differences between the tacrolimus and dupilumab groups, the reader would benefit from key contextual information about the typical treatment journey of AD patients in the Campania Region:

  • Who prescribes these medications (dermatologist, general practitioner)?
  • Where are they dispensed (community pharmacy, hospital pharmacy)?
  • What are the treatment regimens (e.g., twice daily, twice weekly, biweekly injection)?
  • How often must patients renew their prescriptions (ie. visit their doctor/pharmacist)?
  • Who administers the treatment (self-administration vs. healthcare professional)?
  • What are the costs of the medications (reimbursed, out-of-pocket, cost per day)?
  • What do treatment guidelines recommend? (For tacrolimus, as I recall from class, treatment should be stopped or used intermittently once symptoms improve)
  • What criteria guide the prescriber in the choice between tacrolimus ointment and dupilumab injection? (Both products are indicated for other conditions as well.)

The use of prescription data—rather than dispensing data or, ideally, administration data—is a clear limitation, which the authors have rightly acknowledged.

Author Response

Response to Reviewer 3 Comments
1. Summary
Interesting approach using a data-driven method to compare adherence between two distinct treatment plans for atopic dermatitis (AD) patients.
Dear Reviewer, on behalf of all co-authors, we thank you for the valuable comments which we implemented in this new revised version of the manuscript. Please find detailed responses below and the corresponding revisions/corrections in track changes in the resubmitted files.
2. Point-by-point response to Comments and Suggestions for Authors
Comment 1: While the main impact qualifier is described as “pharmaceutical formulation,” it is actually the route of administration (topical vs. subcutaneous injection) that plays a more significant role.
Response 1: We agree that the “route of administration” is more pertinent in our comparative analysis. In response to this suggestion, we have revised the title and carefully reviewed the entire manuscript to ensure greater clarity and consistency. Throughout the revised version, we explicitly refer to the route of administration as the main comparator, while also acknowledging the broader contextual differences between the two pharmacological treatments studied. These include their regulatory classification, prescriber type, dispensing setting, and position in the therapeutic pathway. All relevant details have been revised accordingly, and a new supplementary table (Table S2) has been added to summarize these differences.
Comment 2: To fully understand the differences between the tacrolimus and dupilumab groups, the reader would benefit from key contextual information about the typical treatment journey of AD patients in the Campania Region:
Response 2: Many thanks for these suggestions. In line with these recommendations, we have substantially revised the methodology section (see paragraph 2.2 “Adherence assessment”) to provide a clearer contextualization of the treatment pathway of AD patients in the Campania Region. Specifically, we have detailed the differences between tacrolimus and dupilumab in terms of their regulatory classification, dispensing setting, prescriber type, and clinical indication within the therapeutic algorithm.
Additionally, we have included a new Supplementary Table (Table S2) that summarizes and compares these differences in a structured and concise format. We feel that this will support the reader in understanding the distinct real-world positioning of each treatment within the healthcare delivery system and its potential influence on medication adherence patterns.
Comment 3: Who prescribes these medications (dermatologist, general practitioner)?
Response 3: Both tacrolimus ointment and dupilumab are prescribed by dermatology specialists. Tacrolimus, although dispensed through community pharmacies, is typically initiated and managed by dermatologists. Similarly, dupilumab is prescribed within hospital settings by dermatologists following a formal eligibility assessment for systemic therapy. These prescribing patterns are consistent with current clinical practice in the Campania Region and reflect the regulatory framework and therapeutic indications of both treatments. We now detail this in the revised version of the manuscript (Section 2.2) and in the newly added Supplementary Table S2.
Comment 4: Where are they dispensed (community pharmacy, hospital pharmacy)?
Response 4: Tacrolimus ointment is dispensed through community pharmacies, in accordance
with its Class A reimbursement status within the Italian healthcare system. In contrast, dupilumab is classified as a Class H medication and is dispensed through hospital pharmacies, following approval for systemic treatment in moderate to severe cases of atopic dermatitis. This information is available in the revised version of the manuscript (Section 2.2) and also in Supplementary Tables S1 and S2, as well as in the discussion.
Comment 5: What are the treatment regimens (e.g., twice daily, twice weekly, biweekly injection)?
Response 5: Tacrolimus ointment is typically applied topically twice daily, particularly during flare-ups or induction phases. Dupilumab is administered as a subcutaneous injection every other week (biweekly), following an initial loading dose, in accordance with its approved posology for moderate-to-severe atopic dermatitis. We now detail this in the revised version of the manuscript (Section 2.2) and in the newly added Supplementary Table S2, as well as in the discussion.
Comment 6: How often must patients renew their prescriptions (ie, visit their doctor/pharmacist)?
Response 6: Tacrolimus ointment is dispensed through community pharmacies, and prescriptions can typically be renewed monthly or as needed, depending on the quantity prescribed and the local prescribing habits. Dupilumab, being a hospital-only (Class H) medication, is dispensed exclusively through hospital pharmacies, and prescriptions are generally renewed every two to three months, following regular clinical evaluations by dermatology specialists in hospital-based settings. We now detail this in the revised version of the manuscript (Section 2.2) and in the newly added Supplementary Table S2, as well as in the discussion.
Comment 7: Who administers the treatment (self-administration vs. healthcare professional)?
Response 7: Both tacrolimus and dupilumab are intended for self-administration. Tacrolimus ointment is applied topically by the patient, typically twice daily, while dupilumab is administered as a subcutaneous injection every other week, also by the patient or a caregiver, after appropriate instruction by a healthcare professional. No in-clinic administration is required on a regular basis for either treatment. However, initiation of dupilumab therapy often involves an initial dose administered under medical supervision, followed by self-administration at home. We now detail this in the revised version of the manuscript (Section 2.2) and in the newly added Supplementary Table S2, as well as in the discussion.
Comment 8: What are the costs of the medications (reimbursed, out-of-pocket, cost per day)?
Response 8: As shown in Supplementary Table S1, tacrolimus ointment (0.03%) has an estimated cost of €11.11 per 10g package (public price with VAT included), while dupilumab (300mg/2mL) has a cost of €1,906.54 per syringe. Both medications are fully reimbursed by the Italian National Health Service (NHS), meaning that there is no direct out-of-pocket cost for eligible patients. However, their economic burden on the NHS differs substantially due to differences in unit cost and treatment duration. Specifically, the daily cost of dupilumab is considerably higher compared to tacrolimus, reflecting their distinct mechanisms of action and treatment indications.
To address this important aspect, we have expanded the discussion section in the revised manuscript to include a paragraph on the differential budgetary impact of the two therapies, highlighting how cost considerations may interact with medication adherence and treatment decision-making in the context of chronic disease management.
Comment 9: What do treatment guidelines recommend? (For tacrolimus, as I recall from class,
treatment should be stopped or used intermittently once symptoms improve)
Response 9: Yes, the intermittent or as-needed use of topical tacrolimus once clinical improvement is achieved is explicitly recommended by several clinical guidelines. In particular, the Italian SIDeMaST guidelines (Italian Society of Dermatology and Sexually Transmitted Diseases) recommend the use of tacrolimus ointment for reactive or proactive treatment strategies, emphasizing intermittent maintenance therapy to prevent flare-ups once disease control is achieved. This therapeutic strategy is distinct from continuous maintenance regimens and may lead to lower persistence in prescription records, since patients are recommended to suspend or reduce application when symptoms are under control. We have addressed this important clinical aspect in the revised manuscript (Section 2.2 of materials and methods and discussion section), highlighting that guideline-recommended treatment patterns for tacrolimus may partially explain the shorter persistence duration observed in real-world settings.
Comment 10: What criteria guide the prescriber in the choice between tacrolimus ointment and dupilumab injection? (Both products are indicated for other conditions as well.)
Response 10: The choice between tacrolimus ointment and dupilumab injection in patients with AD is primarily guided by clinical severity, treatment response history, and eligibility for systemic therapy, as defined by national and international guidelines. Hence, the Italian SIDeMaST guidelines recommend topical tacrolimus for patients with moderate to severe AD, typically as an alternative to topical corticosteroids. So, tacrolimus ointment is usually prescribed after failure or contraindication to topical steroids, and before considering systemic treatments.
While, dupilumab is also indicated for patients with moderate to severe AD but for those who are eligible for systemic therapy, particularly those with inadequate disease control despite topical treatment, or who experience adverse effects or contraindications to conventional systemic immunosuppressants (e.g., cyclosporine). Its prescription is thus typically subsequent to failure of topical agents, and is contingent on a structured eligibility assessment often conducted in the hospital setting. This revised version of the manuscript now reflects these differences to better contextualize the adherence analyses.
Comment 11: The use of prescription data—rather than dispensing data or, ideally, administration data—is a clear limitation, which the authors have rightly acknowledged.
Response 11: Many thanks for highlighting this important methodological point. We fully agree that relying on prescription data may represent a limitation in terms of measuring real-world adherence. As acknowledged in the discussion section of the revised manuscript, this limitation may lead to an overestimation of adherence, particularly in the initiation phase, as we cannot guarantee that a prescription was filled or that the medication was taken as intended.
However, this methodological constraint is a common issue in retrospective drug utilization studies based on routinely collected administrative data, especially in the absence of integrated prescription-dispensation-administration systems.

Round 2

Reviewer 1 Report

Comments and Suggestions for Authors

I appreciate the effort and replies but I am still against the original idea of the research and cannot see that comparing two different treatment regimens could lead to valid conclusions.